# MINCUT POOLING IN GRAPH NEURAL NETWORKS

## ABSTRACT

The advance of node pooling operations in Graph Neural Networks (GNNs) has lagged behind the feverish design of new message-passing techniques, and pooling remains an important and challenging endeavor for the design of deep architectures. In this paper, we propose a pooling operation for GNNs that leverages a differentiable unsupervised loss based on the `minCUT` optimization objective. For each node, our method learns a soft cluster assignment vector that depends on the node features, the target inference task (e.g., a graph classification loss), and, thanks to the `minCUT` objective, also on the connectivity structure of the graph. Graph pooling is obtained by applying the matrix of assignment vectors to the adjacency matrix and the node features. The proposed method can also be used as a stand-alone module to cluster vertexes in annotated graphs and solve unsupervised problems. We validate the effectiveness of the proposed pooling method on downstream tasks, including supervised graph classification and a set of unsupervised tasks, which reveal the limitations of existing GNN pooling approaches.

## 1 INTRODUCTION

A fundamental component in deep convolutional neural networks is the *pooling* operation, which replaces the output of convolutions with local summaries of nearby points and is usually implemented by maximum or average operations (Lee et al., 2016). State-of-the-art architectures alternate convolutions, which extrapolate local patterns irrespective of the specific location on the input signal, and pooling, which lets the ensuing convolutions capture aggregated patterns. Pooling allows to learn abstract representations in deeper layers of the network by discarding information that is superfluous for the task, and keeps model complexity under control by limiting the growth of intermediate features.

Graph Neural Networks (GNNs) extend the convolution operation from regular domains, such as images or time series, to data with arbitrary topologies and unordered structures described by graphs (Battaglia et al., 2018). The development of pooling strategies for GNNs, however, has lagged behind the design of newer and more effective message-passing (MP) operations (Gilmer et al., 2017), such as graph convolutions, mainly due to the difficulty of defining an aggregated version of the original graph that supports the pooled signal.

A naïve pooling strategy in GNNs is to average all nodes features (Li et al., 2016), but it has limited flexibility since it does not extract local summaries of the graph structure, and no further MP operations can be applied afterwards. An alternative approach consists in pre-computing coarsened versions of the original graph and then fit the data to these deterministic structures (Bruna et al., 2013). While this aggregation accounts for the connectivity of the graph, it ignores task-specific objectives as well as the node features.

In this paper, we propose a differentiable pooling operation implemented as a neural network layer, which can be seamlessly combined with other MP layers (see Fig. 1). The parameters in the pooling layer are learned by combining the task-specific loss with an unsupervised regularization term, which optimizes a continuous relaxation of the normalized `minCUT` objective. The `minCUT` identifies dense graph components, where the nodes features become locally homogeneous after the message-passing. By gradually aggregating these components, the GNN learns to distil global properties from the graph. The proposed `minCUT` pooling operator (minCUTpool) yields partitions that 1) cluster together nodes which have similar features and are strongly connected on the graph, and 2) take into account the objective of the downstream task.

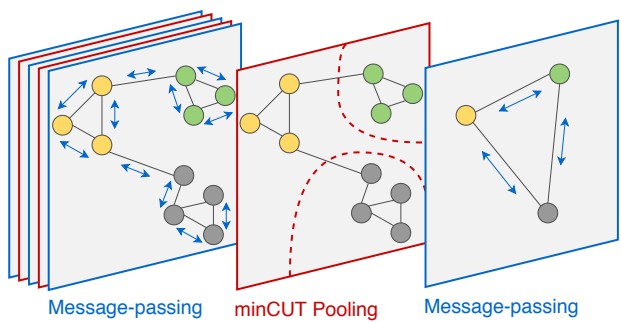

Figure 1: A deep GNN architecture where message-passing is followed by minCUT pooling.

## 2 BACKGROUND

### 2.1 MINCUT AND SPECTRAL CLUSTERING

Given a graph $G = \{\mathcal{V}, \mathcal{E}\}$, $|\mathcal{V}| = N$, and the associated adjacency matrix $\mathbf{A} \in \mathbb{R}^{N \times N}$, the $K$-*way normalized* minCUT (simply referred to as minCUT) aims at partitioning $\mathcal{V}$ in $K$ disjoint subsets by removing the minimum volume of edges. The problem is equivalent to maximizing

$$\frac{1}{K} \sum_{k=1}^{K} \frac{\text{links}(\mathcal{V}_k)}{\text{degree}(\mathcal{V}_k)} = \frac{1}{K} \sum_{k=1}^{K} \frac{\sum_{i,j \in \mathcal{V}_k} \mathcal{E}_{i,j}}{\sum_{i \in \mathcal{V}_k, j \in \mathcal{V} \setminus \mathcal{V}_k} \mathcal{E}_{i,j}}, \tag{1}$$

where the numerator counts the edge volume within each cluster, and the denominator counts the edges between the nodes in a cluster and the rest of the graph (Shi & Malik, 2000). Let $\mathbf{C} \in \mathbb{R}^{N \times K}$ be a *cluster assignment matrix*, so that $\mathbf{C}_{i,j} = 1$ if node $i$ belongs to cluster $j$, and 0 otherwise. The minCUT problem can be expressed as

$$\text{maximize} \quad \frac{1}{K} \sum_{k=1}^{K} \frac{\mathbf{C}_k^T \mathbf{A} \mathbf{C}_k}{\mathbf{C}_k^T \mathbf{D} \mathbf{C}_k}, \quad \text{s.t.} \ \ \mathbf{C} \in \{0, 1\}^{N \times K}, \ \ \mathbf{C} \mathbf{1}_K = \mathbf{1}_N, \tag{2}$$

where $\mathbf{D} = \text{diag}(\mathbf{A} \mathbf{1}_N)$ is the degree matrix (Dhillon et al., 2004). Since problem (2) is NP-hard, it is usually recast in a relaxed formulation that can be solved in polynomial time and guarantees a near-optimal solution (Yu & Shi, 2003):

$$\underset{\mathbf{Q} \in \mathbb{R}^{N \times K}}{\arg \max} \quad \frac{1}{K} \sum_{k=1}^{K} \mathbf{Q}_k^T \mathbf{A} \mathbf{Q}_k, \quad \text{s.t.} \ \mathbf{Q} = \mathbf{C}(\mathbf{C}^T \mathbf{D} \mathbf{C})^{-\frac{1}{2}}, \ \mathbf{Q}^T \mathbf{Q} = \mathbf{I}_K. \tag{3}$$

While the optimization problem (3) is still non-convex, there exists an optimal solution $\mathbf{Q}^* = \mathbf{U}_K \mathbf{O}$, where $\mathbf{U}_K \in \mathbb{R}^{N \times K}$ contains the eigenvectors of $\mathbf{A}$ corresponding to the $K$ largest eigenvalues, and $\mathbf{O} \in \mathbb{R}^{K \times K}$ is an orthogonal transformation (Ikebe et al., 1987).

Since the elements of $\mathbf{Q}^*$ are real values rather than binary cluster indicators, the *spectral clustering* (SC) approach can be used to find discrete cluster assignments. In SC, the rows of $\mathbf{Q}^*$ are treated as node representations embedded in the eigenspace of the Laplacian, and are clustered together with standard algorithms such as $k$-means (Von Luxburg, 2007). One of the main limitations of SC lies in the computation of the spectrum of $\mathbf{A}$, which has a memory complexity of $\mathcal{O}(N^2)$ and a computational complexity of $\mathcal{O}(N^3)$. This prevents its applicability to large datasets.

To deal with such scalability issues, the constrained optimization in (3) can be solved by gradient descent algorithms that refine the solution by iterating operations whose individual complexity is $\mathcal{O}(N^2)$, or even $\mathcal{O}(N)$ (Han & Filippone, 2017). Those algorithms search the solution on the manifold induced by the orthogonality constraint on the columns of $\mathbf{Q}$, by performing gradient updates along the geodesics (Wen & Yin, 2013; Collins et al., 2014). Alternative approaches rely on the QR factorization to constrain the space of feasible solutions (Damle et al., 2016), and alleviate the cost $\mathcal{O}(N^3)$ of the factorization by ensuring that orthogonality holds only on one minibatch at a time (Shaham et al., 2018).

Other works based on neural networks include an autoencoder trained to map the $i$th row of the Laplacian to the $i$th components of the first $K$ eigenvectors, to avoid the spectral decomposition (Tian et al., 2014). Yi et al. (2017) use a soft orthogonality constraint to learn spectral embeddings as a volumetric reparametrization of a precomputed Laplacian eigenbase. Shaham et al. (2018); Kampffmeyer et al. (2019) propose differentiable loss functions to partition generic data and process out-of-sample data at inference time. Nazi et al. (2019) generate balanced node partitions with a GNN, but adopt an optimization that does not encourage cluster assignments to be orthogonal.

## 2.2 GRAPH NEURAL NETWORKS

Many approaches have been proposed to process graphs with neural networks, including recurrent architectures (Scarselli et al., 2009; Li et al., 2016) or convolutional operations inspired by filters used in graph signal processing (Defferrard et al., 2016; Bianchi et al., 2019). Since our focus is on graph pooling, we base our GNN implementation on a simple MP operation, which combines the features of each node with its 1st-order neighbors. To account for the initial node features, it is possible to introduce self-loops by adding a (scaled) identity matrix to the diagonal of $\mathbf{A}$ (Kipf & Welling, 2017). Since our pooling will modify the structure of the adjacency matrix, we prefer a MP implementation that leaves the original $\mathbf{A}$ unaltered and accounts for the initial node features by means of skip connections.

Let $\tilde{\mathbf{A}} = \mathbf{D}^{-\frac{1}{2}}\mathbf{A}\mathbf{D}^{-\frac{1}{2}} \in \mathbb{R}^{N \times N}$ be the symmetrically normalized adjacency matrix and $\mathbf{X} \in \mathbb{R}^{N \times F}$ the matrix containing the node features. The output of the MP layer is

$$\mathbf{X}^{(t+1)} = MP(\mathbf{X}^{(t)}, \tilde{\mathbf{A}}) = \text{ReLU}(\tilde{\mathbf{A}}\mathbf{X}^{(t)}\mathbf{W}_m + \mathbf{X}^{(t)}\mathbf{W}_s), \tag{4}$$

where $\boldsymbol{\Theta}_{MP} = \{\mathbf{W}_m, \mathbf{W}_s\}$ are the trainable weights relative to the mixing and skip component of the layer, respectively.

## 3 PROPOSED METHOD

The minCUT pooling strategy computes a cluster assignment matrix $\mathbf{S} \in \mathbb{R}^{N \times K}$ by means of a multi-layer perceptron, which maps each node feature $\mathbf{x}_i$ into the $i$th row of $\mathbf{S}$:

$$\mathbf{S} = softmax(\text{ReLU}(\mathbf{X}\mathbf{W}_1)\mathbf{W}_2), \tag{5}$$

where $\boldsymbol{\Theta}_{Pool} = \{\mathbf{W}_1 \in \mathbb{R}^{F \times H}, \mathbf{W}_2 \in \mathbb{R}^{H \times K}\}$ are trainable parameters. The *softmax* function guarantees that $s_{i,j} \in [0, 1]$ and enforces the constraints $\mathbf{S}\mathbf{1}_K = \mathbf{1}_N$ inherited from the optimization problem in (2). The parameters $\boldsymbol{\Theta}_{MP}$ and $\boldsymbol{\Theta}_{Pool}$ are jointly optimized by minimizing the usual task-specific loss, as well as an unsupervised loss $\mathcal{L}_u$, which is composed of two terms

$$\mathcal{L}_u = \mathcal{L}_c + \mathcal{L}_o = \underbrace{-\frac{Tr(\mathbf{S}^T\tilde{\mathbf{A}}\mathbf{S})}{Tr(\mathbf{S}^T\tilde{\mathbf{D}}\mathbf{S})}}_{\mathcal{L}_c} + \underbrace{\left\| \frac{\mathbf{S}^T\mathbf{S}}{\|\mathbf{S}^T\mathbf{S}\|_F} - \frac{\mathbf{I}_K}{\sqrt{K}} \right\|_F}_{\mathcal{L}_o}, \tag{6}$$

where $\|\cdot\|_F$ indicates the Frobenius norm.

The cut loss term, $\mathcal{L}_c$, evaluates the `minCUT` given by the cluster assignment $\mathbf{S}$, and is bounded by $-1 \leq \mathcal{L}_c \leq 0$. Minimizing $\mathcal{L}_c$ encourages strongly connected nodes to be clustered together, since the inner product $\langle \mathbf{s}_i, \mathbf{s}_j \rangle$ increases when $\tilde{a}_{i,j}$ is large. $\mathcal{L}_c$ has a single maximum, reached when the numerator $Tr(\mathbf{S}^T\tilde{\mathbf{A}}\mathbf{S}) = \frac{1}{K}\sum_{k=1}^{K}\mathbf{S}_k^T\tilde{\mathbf{A}}\mathbf{S}_k = 0$. This occurs if, for each pair of connected nodes (i.e., $\tilde{a}_{i,j} > 0$), the cluster assignments are orthogonal (i.e., $\langle \mathbf{s}_i, \mathbf{s}_j \rangle = 0$). $\mathcal{L}_c$ reaches its minimum, $-1$, when $Tr(\mathbf{S}^T\tilde{\mathbf{A}}\mathbf{S}) = Tr(\mathbf{S}^T\tilde{\mathbf{D}}\mathbf{S})$. This occurs when in a graph with $K$ disconnected components the cluster assignments are equal for all the nodes in the same component and orthogonal to the cluster assignments of nodes in different components. However, $\mathcal{L}_c$ is a non-convex function and its minimization can lead to local minima or degenerate solutions. For example, given a connected graph, a trivial optimal solution is the one that assigns all nodes to the same cluster. As a consequence of the continuous relaxation, another degenerate minimum occurs when the cluster assignments are all uniform, that is, all nodes are equally assigned to all clusters. This problem is exacerbated by prior message-passing operations, which make the node features more uniform.

The orthogonality loss term, $\mathcal{L}_o$, penalizes the degenerate minima of $\mathcal{L}_c$ by encouraging the cluster assignments to be orthogonal and the clusters to be of similar size. Since the two matrices in $\mathcal{L}_o$ have unitary norm it is easy to see that $0 \leq \mathcal{L}_o \leq 2$. Therefore, $\mathcal{L}_o$ does not dominate over $\mathcal{L}_c$ and the two terms can be safely summed directly (see Fig. 4 for an example). $\mathbf{I}_K$ can be interpreted as a (rescaled) clustering matrix $\mathbf{I}_K = \hat{\mathbf{S}}^T \hat{\mathbf{S}}$, where $\hat{\mathbf{S}}$ assigns exactly $N/K$ points to each cluster. The value of the Frobenius norm between clustering matrices is not dominated by the performance on the largest clusters (Law et al., 2017) and, thus, can be used to optimize intra-cluster variance.

Contrarily to SC methods that search for feasible solutions only within the space of orthogonal matrices, $\mathcal{L}_o$ only introduces a soft constraint that could be violated during the learning procedure. Since $\mathcal{L}_c$ is non-convex, the violation compromises the theoretical guarantee of convergence to the optimum of (3). However, we note that:

1. the cluster assignments $\mathbf{S}$ are well initialized: after the MP operation, the features of the connected vertices become similar and, since the MLP is a smooth function (Nelles, 2013), it yields similar cluster assignments for those vertices;

2. in the GNN architecture, the `minCUT` objective is a regularization term and, therefore, a solution which is sub-optimal for (3) could instead be adequate for the specific objective of the downstream task;

3. optimizing the task-specific loss helps the GNN to avoid the degenerate minima of $\mathcal{L}_c$.

## 3.1 COARSENING

The coarsened version of the adjacency matrix and the graph signal are computed as

$$\mathbf{A}^{pool} = \mathbf{S}^T \tilde{\mathbf{A}} \mathbf{S}; \quad \mathbf{X}^{pool} = \mathbf{S}^T \mathbf{X}, \tag{7}$$

where the entry $x_{i,j}^{pool}$ in $\mathbf{X}^{pool} \in \mathbb{R}^{K \times F}$ is the weighted average value of feature $j$ among the elements in cluster $i$. $\mathbf{A}^{pool} \in \mathbb{R}^{K \times K}$ is a symmetric matrix, whose entries $a_{i,i}^{pool}$ are the total number of edges between the nodes in the cluster $i$, while $a_{i,j}^{pool}$ is the number of edges between cluster $i$ and $j$. Since $\mathbf{A}^{pool}$ corresponds to the numerator of $\mathcal{L}_c$ in (7), the trace maximization yields clusters with many internal connections and weakly connected to each other. Hence, $\mathbf{A}^{pool}$ will be a diagonal-dominant matrix, which describes a graph with self-loops much stronger than any other connection. Because self-loops hamper the propagation across adjacent nodes in the MP operations following the pooling layer, we compute the new adjacency matrix $\tilde{\mathbf{A}}^{pool}$ by zeroing the diagonal and by applying the degree normalization

$$\hat{\mathbf{A}} = \mathbf{A}^{pool} - \mathbf{I}_K diag(\mathbf{A}^{pool}); \quad \tilde{\mathbf{A}}^{pool} = \hat{\mathbf{D}}^{-\frac{1}{2}} \hat{\mathbf{A}} \hat{\mathbf{D}}^{-\frac{1}{2}}. \tag{8}$$

where $diag(\cdot)$ returns the matrix diagonal.

## 3.2 DISCUSSION AND RELATIONSHIP WITH SPECTRAL CLUSTERING

The proposed method is straightforward to implement: the cluster assignments, the loss, graph coarsening, and feature pooling are all computed with standard linear algebra operations.

There are several differences between minCUTpool and classic SC methods. SC partitions the graph based on the Laplacian, but does not account for the node features. Instead, the cluster assignments $\mathbf{s}_i$ found by minCUTpool depend on $\mathbf{x}_i$, which works well if connected nodes have similar features. This is a reasonable assumption in GNNs since, even in disassortative graphs (i.e., networks where dissimilar nodes are likely to be connected (Newman, 2003)), the features tend to become similar due to the MP operations.

Another difference is that SC handles a single graph and is not conceived for tasks with multiple graphs to be partitioned independently. Instead, thanks to the independence of the model parameters from the number of nodes $N$ and from the graph spectrum, minCUTpool can generalize to out-of-sample data. This feature is fundamental in problems such as graph classification, where each sample is a graph with a different structure, and allows to train the model on small graphs and process larger ones at inference time. Finally, minCUTpool directly uses the soft cluster assignments rather than performing $k$-means afterwards.

## 4 RELATED WORK ON POOLING IN GNNS

**Trainable pooling methods.** Similarly to our method, these approaches learn how to generate coarsened version of the graph through differentiable functions, which take as input the nodes features $\mathbf{X}$ and are parametrized by weights optimized on the task at hand.

*Diffpool* (Ying et al., 2018) is a pooling module that includes two parallel MP layers: one to compute the new node features $\mathbf{X}^{(t+1)}$ and another to generate the cluster assignments $\mathbf{S}$. Diffpool implements an unsupervised loss that consists of two terms. First, the *link prediction* term $\|\mathbf{A} - \mathbf{S}\mathbf{S}^T\|_F$ minimizes the Frobenius norm of the difference between the adjacency and the Gram matrix of the cluster assignments, encouraging nearby nodes to be clustered together. The second term $\frac{1}{N}\sum_{i=1}^{N} H(\mathbf{S}_i)$ minimizes the entropy of the cluster assignments to make them alike to one-hot vectors. Like minCUTpool, Diffpool clusters the vertices of annotated graphs, but yields completely different partitions, since it computes differently the clustering assignments, the coarsened adjacency matrix and, most importantly, the unsupervised loss. In Diffpool, such a loss shows pathological behaviors that are discussed later in the experiments.

The approach dubbed *Top-K* pooling (Hongyang Gao, 2019; Lee et al., 2019), learns a projection vector that is applied to each node feature to obtain a score. The nodes with the $K$ highest scores are retained, the others are dropped. Since the top-$K$ selection is not differentiable, the scores are also used as a gate/attention for the node features, letting the projection vector to be trained with backpropagation. Top-$K$ is memory efficient as it avoids generating cluster assignments. To prevent $\mathbf{A}$ from becoming disconnected after nodes removal, Top-$K$ drops the rows and the columns from $\mathbf{A}^2$ and uses it as the new adjacency matrix. However, computing $\mathbf{A}^2$ costs $\mathcal{O}(N^2)$ and it is inefficient to implement with sparse operations.

**Topological pooling methods.** These methods pre-compute a pyramid of coarsened graphs, only taking into account the topology ($\mathbf{A}$), but not the node features ($\mathbf{X}$). During training, the node features are pooled with standard procedures and are fit into these deterministic graph structures. These methods are less flexible, but provide a stronger bias that can prevent degenerate solutions (e.g., coarsened graphs collapsing in a single node).

The approach proposed by Bruna et al. (2013), which has been adopted also in other GNN architectures (Defferrard et al., 2016; Fey et al., 2018), exploits *GRACLUS* (Dhillon et al., 2004), a hierarchical algorithm based on SC. At each pooling level $l$, GRACLUS indetifies the pairs of maximally similar nodes $i_l$ and $j_l$ to be clustered together into a new vertex $k_{(l+1)}$. At inference phase, max-pooling is used to determine which node in the pair is kept. Fake vertices are added so that the number of nodes can be halved each time, but this injects noisy information in the graph.

*Node decimation* is a method originally proposed in graph signal processing literature (Shuman et al., 2016), which as been adapted also for GNNs (Simonovsky & Komodakis, 2017). The nodes are partitioned in two sets, according to the signs of the Laplacian eigenvector associated to the largest eigenvalue. One of the two sets is dropped, reducing the number of nodes each time approximately by half. Kron reduction is used to compute a pyramid of coarsened Laplacians from the remaining nodes.

A procedure proposed in Gama et al. (2018) diffuses a signal from designated nodes on the graph and stores the observed sequence of diffused components. The resulting stream of information is interpreted as a time signal, where standard CNN pooling is applied. We also mention a pooling operation for coarsening binary unweighted graphs by aggregating maximal cliques (Luzhnica et al., 2019). Nodes assigned to the same clique are summarized by max or average pooling and become a new node in the coarsened graph.

## 5 EXPERIMENTS

We consider both supervised and unsupervised tasks, and compare minCUTpool with other GNN pooling strategies. The Appendix provides further details on the experiments and a schematic depiction of the architectures used in each task. In addition, the Appendix reports two additional experiments: i) graph reconstruction by means of an Auto Encoder with bottleneck, implemented with pooling and un-pooling layers, ii) an architecture with pooling for graph regression.

## 5.1 CLUSTERING THE GRAPH NODES

To study the effectiveness of the proposed loss, we perform different node clustering tasks with a simple GNN composed of a single MP layer followed by a pooling layer. The GNN is trained by minimizing $\mathcal{L}_u$ only, so that its effect is evaluated without the "interference" of a supervised loss.

**Clustering on synthetic networks**   We consider two simple graphs: the first is a network with 6 communities and the second is a regular grid. The adjacency matrix $\mathbf{A}$ is binary and the features $\mathbf{X}$ are the 2-D node coordinates. Fig. 2 depicts the node partitions generated by SC (a, d), Diffpool (b, e), and minCUTpool (c, f). Cluster indexes for Diffpool and minCUTpool are obtained by taking the argmax of $\mathbf{S}$ row-wise. Compared to SC, Diffpool and minCUTpool leverage the information contained in $\mathbf{X}$. minCUTpool generates very accurate and balanced partitions, demonstrating that the cluster assignment matrix $\mathbf{S}$ is well formed. On the other hand, Diffpool assigns some nodes to the wrong community in the first example, and produces an imbalanced partition of the grid.

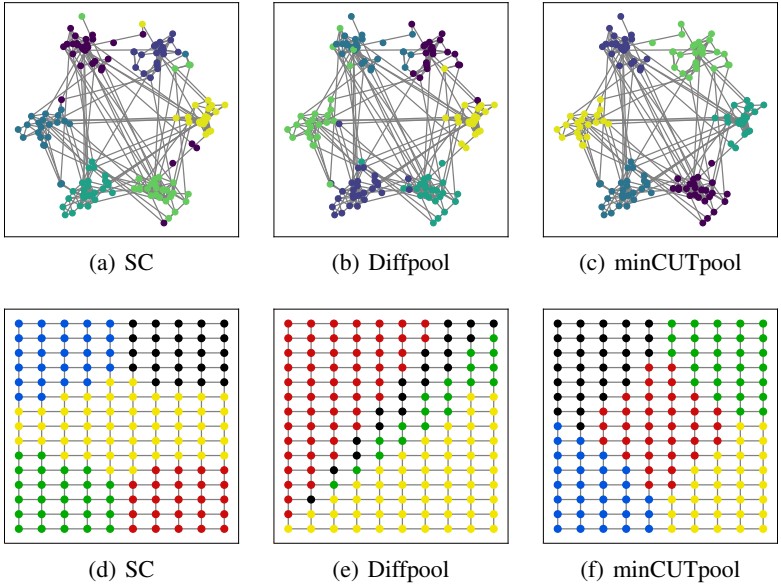

(a) SC       (b) Diffpool       (c) minCUTpool

(d) SC       (e) Diffpool       (f) minCUTpool

Figure 2: Node clustering on a community network ($K$=6) and on a grid graph ($K$=5).

**Image segmentation**   Given an image, we build a Region Adjacency Graph (Trémeau & Colantoni, 2000) using as nodes the regions generated by an oversegmentation procedure (Felzenszwalb & Huttenlocher, 2004). The SC technique used in this example is the recursive normalized cut (Shi & Malik, 2000), which recursively clusters the nodes until convergence. For Diffpool and min-CUTpool, we include node features consisting of the average and total color in each oversegmented region. We set the number of desired clusters to $K = 4$. The results in Fig. 3 show that minCUTpool yields a more precise segmentation. On the other hand, SC and Diffpool aggregate wrong regions and, in addition, SC finds too many segments.

**Clustering on citation networks**   We cluster the nodes of three popular citation networks: Cora, Citeseer, and Pubmed. The nodes are documents represented by sparse bag-of-words feature vectors stored in $\mathbf{X}$ and the binary undirected edges in $\mathbf{A}$ indicate citation links between documents. Each node $i$ is labeled with the document class $y_i$. Once the training is over, to test the quality of the partitions generated by each method we check the agreement between the cluster assignments and the true class labels. Tab. 1 reports the Completeness Score $\text{CS}(\tilde{\mathbf{y}}, \mathbf{y}) = 1 - \frac{H(\tilde{\mathbf{y}}|\mathbf{y})}{H(\tilde{\mathbf{y}})}$ and Normalized Mutual Information $\text{NMI}(\tilde{\mathbf{y}}, \mathbf{y}) = \frac{H(\tilde{\mathbf{y}}) - H(\tilde{\mathbf{y}}|\mathbf{y})}{\sqrt{H(\tilde{\mathbf{y}}) - H(\mathbf{y})}}$, where $H(\cdot)$ is the entropy.

The GNN architecture configured with minCUTpool achieves a higher NMI score than SC, which does not account for the node features $\mathbf{X}$ when generating the partitions. Our pooling operation

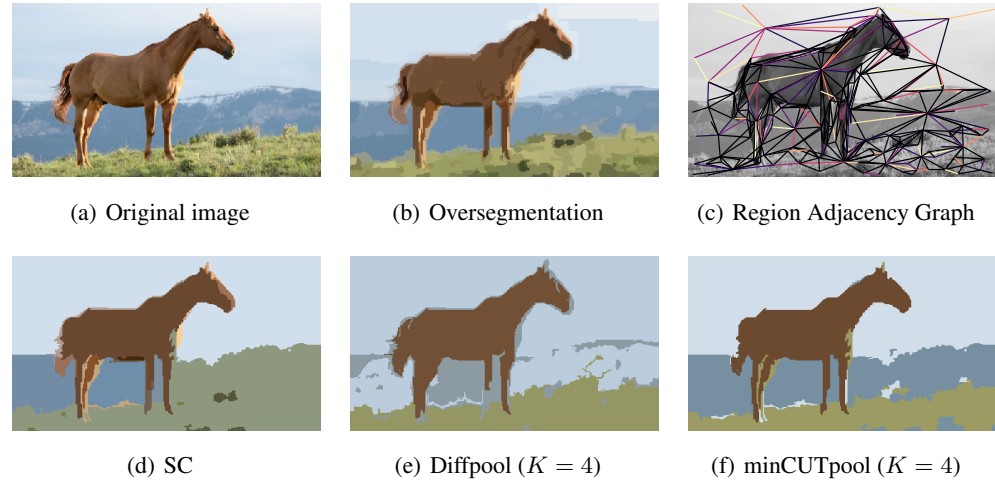

Figure 3: Image segmentation by clustering the nodes of the Region Adjacency Graph.

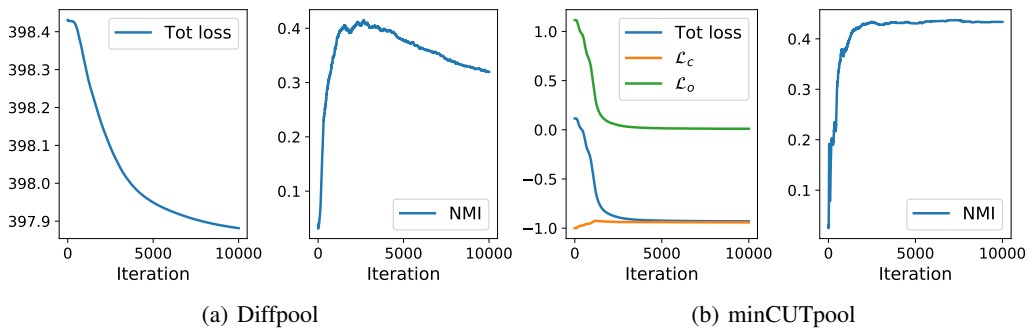

Figure 4: Unsupervised losses and NMI of Diffpool and minCUTpool on Cora.

outperforms also Diffpool, since the minimization of the unsupervised loss in Diffpool yields degenerate solutions. The pathological behavior is shown in Fig. 4, which depicts the evolution of the NMI scores as the unsupervised losses in Diffpool and minCUTpool are minimized in training.

Table 1: NMI and CS obtained by clustering the nodes on citation networks over 10 different runs. The number of clusters $K$ is equal to the number of node classes.

| Dataset | $K$ | Spectral clustering | | Diffpool | | minCUTpool | |
|---|---|---|---|---|---|---|---|
| | | NMI | CS | NMI | CS | NMI | CS |
| Cora | 7 | $0.025 \pm 0.014$ | $0.126 \pm 0.042$ | $0.315 \pm 0.005$ | $0.309 \pm 0.005$ | $\mathbf{0.404} \pm 0.018$ | $\mathbf{0.392} \pm 0.018$ |
| Citeseer | 6 | $0.014 \pm 0.003$ | $0.033 \pm 0.000$ | $0.139 \pm 0.016$ | $0.153 \pm 0.020$ | $\mathbf{0.287} \pm 0.047$ | $\mathbf{0.283} \pm 0.046$ |
| Pubmed | 3 | $0.182 \pm 0.000$ | $\mathbf{0.261} \pm 0.000$ | $0.079 \pm 0.001$ | $0.085 \pm 0.001$ | $\mathbf{0.200} \pm 0.020$ | $0.197 \pm 0.019$ |

## 5.2 SUPERVISED GRAPH CLASSIFICATION

In this task, the $i$-th datum is a graph with $N_i$ nodes represented by a pair $\{\mathbf{A}_i, \mathbf{X}_i\}$ and must be associated to the correct label $\mathbf{y}_i$. We test the models on different graph classification datasets. For featureless graphs, we used the node degree information and the clustering coefficient as surrogate node features. We evaluate model performance with a 10-fold train/test split, using $10\%$ of the training set in each fold as validation for early stopping. We adopt a fixed network architecture, *MP(32)-pool-MP(32)-pool-MP(32)-GlobalAvgPool-softmax*, where MP is the message-passing operation in (4)

with 32 hidden units. The pooling module is implemented either by Graclus, Decimation pooling, Top-$K$, SAGPool (Lee et al., 2019), Diffpool, or the proposed minCUTpool. Each pooling method is configured to drop half of the nodes in a graph ($K = N/2$ in Top-$K$, Diffpool, and minCUTpool). As baselines, we consider the popular Weisfeiler-Lehman (WL) graph kernel (Shervashidze et al., 2011), a network with only MP layers (*Flat*), and a fully connected network (*Dense*).

Table 2: Graph classification accuracy. Significantly better results ($p < 0.05$) are in bold.

| Dataset | WL | Dense | Flat | Graclus | Decim. | Diffpool | Top-$K$ | SAGpool | minCUT |
|---|---|---|---|---|---|---|---|---|---|
| Bench-easy | 92.6 | $29.3_{\pm0.3}$ | $98.5_{\pm0.3}$ | $97.5_{\pm0.5}$ | $97.9_{\pm0.5}$ | $98.6_{\pm0.4}$ | $82.4_{\pm8.9}$ | $84.2_{\pm2.3}$ | $\mathbf{99.0_{\pm0.0}}$ |
| Bench-hard | 60.0 | $29.4_{\pm0.3}$ | $67.6_{\pm2.8}$ | $69.0_{\pm1.5}$ | $\mathbf{72.6_{\pm0.9}}$ | $69.9_{\pm1.9}$ | $42.7_{\pm15.2}$ | $37.7_{\pm14.5}$ | $\mathbf{73.8_{\pm1.9}}$ |
| Mutagenicity | $\mathbf{81.7_{\pm1.1}}$ | $68.4_{\pm0.3}$ | $78.0_{\pm1.3}$ | $74.4_{\pm1.8}$ | $77.8_{\pm2.3}$ | $77.6_{\pm2.7}$ | $71.9_{\pm3.7}$ | $72.4_{\pm2.4}$ | $79.9_{\pm2.1}$ |
| Proteins | $71.2_{\pm2.6}$ | $68.7_{\pm3.3}$ | $72.6_{\pm4.8}$ | $68.6_{\pm4.6}$ | $73.3_{\pm3.7}$ | $72.7_{\pm3.8}$ | $69.6_{\pm3.5}$ | $70.5_{\pm2.6}$ | $\mathbf{76.5_{\pm2.6}}$ |
| DD | $78.6_{\pm2.7}$ | $70.6_{\pm5.2}$ | $76.8_{\pm1.5}$ | $70.5_{\pm4.8}$ | $72.0_{\pm3.1}$ | $\mathbf{79.3_{\pm2.4}}$ | $69.4_{\pm7.8}$ | $71.5_{\pm4.5}$ | $\mathbf{80.8_{\pm2.3}}$ |
| COLLAB | $74.8_{\pm1.3}$ | $79.3_{\pm1.6}$ | $\mathbf{82.1_{\pm1.8}}$ | $77.1_{\pm2.1}$ | $79.1_{\pm1.5}$ | $81.8_{\pm1.4}$ | $79.3_{\pm1.8}$ | $79.2_{\pm2.0}$ | $\mathbf{83.4_{\pm1.7}}$ |
| Reddit-Binary | $68.2_{\pm1.7}$ | $48.5_{\pm2.6}$ | $80.3_{\pm2.6}$ | $79.2_{\pm0.4}$ | $84.3_{\pm2.4}$ | $86.8_{\pm2.1}$ | $74.7_{\pm4.5}$ | $73.9_{\pm5.1}$ | $\mathbf{91.4_{\pm1.5}}$ |

Tab. 2 reports the classification results, highlighting those that are significantly better ($p$-value $< 0.05$ w.r.t. the method with the highest mean accuracy). The comparison with *Flat* helps to understand if a pooling operation is useful or not. The results of *Dense*, instead, help to quantify how much additional information is brought by the graph structure, with respect to the node features alone. It can be seen that minCUTpool obtains always equal or better results with respect to every other GNN architecture. On the other hand, some pooling procedures do not always improve the performance compared to the *Flat* baseline, making them not advisable to use in some cases. The WL kernel generally performs worse than the GNNs, except for the Mutagenicity dataset. This is probably because Mutagenicity has smaller graphs than the other datasets, and the adopted GNN architecture is overparametrized for this task. Interestingly, in some dataset such as Proteins and COLLAB it is possible to obtain fairly good classification accuracy with the *Dense* architecture, meaning that the graph structure only adds limited information.

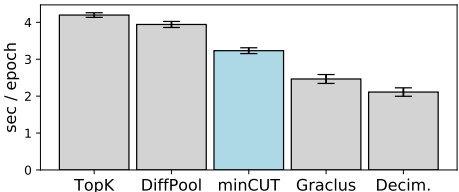

Figure 5: Average duration of one epoch using the same GNN with different pooling operations. Times were computed with an Nvidia GeForce GTX 1050, on the DD dataset with batch size of 1.

Fig. 5 reports a comparison of the execution time per training epoch for each pooling algorithm. Graclus and Decimation are understandably the fastest methods, since the coarsened graphs are pre-computed. Among the differentiable pooling methods, minCUTpool is faster than Diffpool, which uses a slower MP layer rather than a MLP to compute cluster assignments, and than Top-$K$, which computes the square of $\mathbf{A}$ at every forward pass.

## 6 CONCLUSIONS

We proposed a pooling layer for GNNs that coarsens a graph by taking into account both the the connectivity structure and the node features. The layer optimizes a regularization term based on the `minCUT` objective, which is minimized in conjunction with the task-specific loss to produce node partitions that are optimal for the task at hand.

We tested the effectiveness of our pooling strategy on unsupervised node clustering tasks, by optimizing only the unsupervised clustering loss, as well as supervised graph classification tasks on several popular benchmark datasets. Results show that minCUTpool performs significantly better than existing pooling strategies for GNNs.

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

APPENDIX

# A    ADDITIONAL EXPERIMENTS

## A.1    GNN AUTOENCODER

To compare the amount of information retained by the pooling layers in the coarsened graphs, we train an autoencoder (AE) to reconstruct a input graph signal $\mathbf{X}$ from its pooled version. The AE architecture is *MP(32)-MP(32)-pool-unpool-MP(32)-MP(32)-MP*, and is trained by minimizing the mean squared error between the original and the reconstructed graph signal, $\|\mathbf{X} - \mathbf{X}^{\text{rec}}\|^2$. All the pooling operations are configured to retain $25\%$ of the original nodes.

In Diffpool and minCUTpool, the *unpool* step is simply implemented by transposing the original pooling operations

$$\mathbf{X}^{\text{rec}} = \mathbf{S}\mathbf{X}^{\text{pool}}; \quad \mathbf{A}^{\text{rec}} = \mathbf{S}\mathbf{A}^{\text{pool}}\mathbf{S}^T. \tag{9}$$

Top-$K$ does not generate a cluster assignment matrix, but returns a binary mask $\mathbf{m} = \{0, 1\}^N$ that indicates the nodes to drop (0) or to retain (1). Therefore, an upsamplig matrix $\mathbf{U}$ is built by dropping the columns of the identity matrix $\mathbf{I}_N$ that correspond to a 0 in $\mathbf{m}$, $\mathbf{U} = [\mathbf{I}_N]_{:,\mathbf{m}==1}$. The unpooling operation is performed by replacing $\mathbf{S}$ with $\mathbf{U}$ in (9), and the resulting upscaled graph is a version of the original graph with zeroes in correspondence of the dropped nodes.

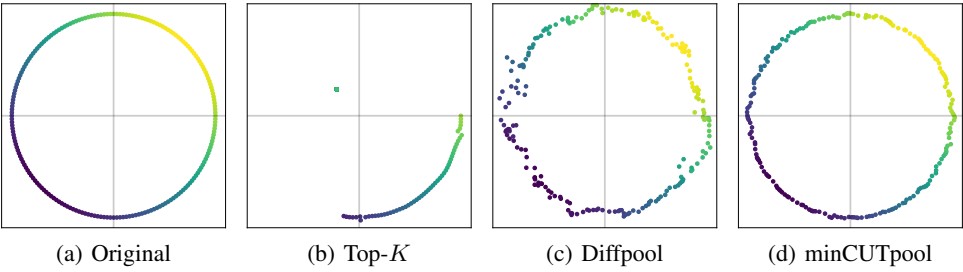

(a) Original          (b) Top-$K$          (c) Diffpool          (d) minCUTpool

Figure 6: AE reconstruction of a ring graph

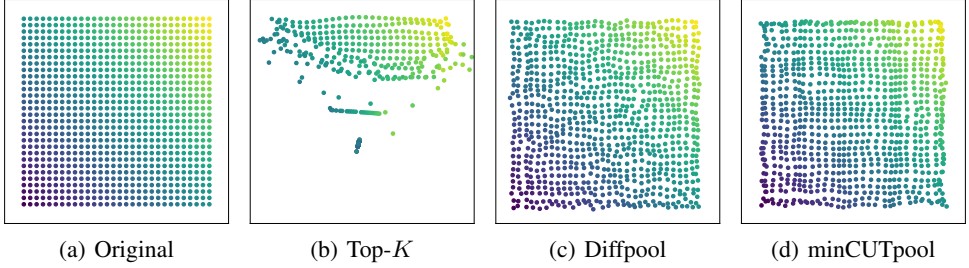

(a) Original          (b) Top-$K$          (c) Diffpool          (d) minCUTpool

Figure 7: AE reconstruction of a grid graph

Fig. 6 and 7 report the original graph signal $\mathbf{X}$ (the node features are the 2-D coordinates of the nodes) and the reconstruction $\mathbf{X}^{\text{rec}}$ obtained by using the different pooling methods, for a ring graph and a regular grid graph. The reconstruction produced by Diffpool is worse for the ring graph, but is almost perfect for the grid graph, while minCUTpool yields good results in both cases. On the other hand, Top-$K$ clearly fails in generating a coarsened representation that maintains enough information from the original graph.

This experiment highlights a major issue in Top-$K$ pooling, which retains the nodes associated to the highest $K$ values of a score vector $\mathbf{s}$, computed by projecting the node features onto a trainable vector $\mathbf{p}$: $\mathbf{s} = \mathbf{X}\mathbf{p}$. Nodes that are connected on the graph usually share similar features, and

their similarity further increases after the MP operations, which combine the features of neighboring nodes. Retaining the nodes associated to the top $K$ scores in $\mathbf{s}$ corresponds to keeping those nodes that are alike and highly connected, as it can be seen in Fig. 6-7. Therefore, Top-$K$ discards entire portions of the graphs, which might contain important information. This explains why Top-$K$ fails to recover the original graph signal when used as bottleneck for the AE, and yields the worse performance among all GNN methods in the graph classification task.

## A.2 GRAPH REGRESSION OF MOLECULAR PROPERTIES ON QM9

The QM9 chemical database is a collection of $\approx$135k small organic molecules, associated to continuous labels describing several geometric, energetic, electronic, and thermodynamic properties[1]. Each molecule in the dataset is represented as a graph $\{\mathbf{A}_i, \mathbf{X}_i\}$, where atoms are associated to nodes, and edges represent chemical bonds. The atomic number of each atom (one-hot encoded; C, N, F, O) is taken as node feature and the type of bond (one-hot encoded; single, double, triple, aromatic) can be used as edge attribute. In this experiment, we ignore the edge attributes in order to use all pooling algorithms without modifications.

The purpose of this experiment is to compare the trainable pooling methods also on a graph regression task, but it must be intended as a proof of concept. In fact, the graphs in this dataset are extremely small (the average number of nodes is 8) and, therefore, a pooling operation is arguably not necessary. We consider a GNN with architecture *MP(32)-pool-MP(32)-GlobalAvgPool-Dense*, where *pool* is implemented by Top-$K$, Diffpool, or minCUTpool. The network is trained to predict a given chemical property from the input molecular graphs. Performance is evaluated with a 10-fold cross-validation, using $10\%$ of the training set for validation in each split. The GNNs are trained for 50 epochs, using Adam with learning rate 5e-4, batch size 32, and ReLU activations. We use the mean squared error (MSE) as supervised loss.

The MSE obtained on the prediction of each property for different pooling methods is reported in Tab. 3. As expected, the flat baseline with no pooling operation (*MP(32)-MP(32)-GlobalAvgPool-Dense*) yields a lower error in most cases. Contrarily to the graph classification and the AE task, Top-$K$ achieves better results than Diffpool in average. Once again, minCUTpool significantly outperforms the other methods on each regression task and, in one case, also the flat baseline.

| Property | Top-$K$ | Diffpool | minCUTpool | Flat baseline |
|---|---|---|---|---|
| mu | $0.600_{\pm 0.085}$ | $0.651_{\pm 0.026}$ | $\mathbf{0.538}_{\pm 0.012}$ | $0.559_{\pm 0.007}$ |
| alpha | $0.197_{\pm 0.087}$ | $0.114_{\pm 0.001}$ | $\underline{0.078}_{\pm 0.007}$ | $\mathbf{0.065}_{\pm 0.006}$ |
| homo | $0.698_{\pm 0.102}$ | $0.712_{\pm 0.015}$ | $\underline{0.526}_{\pm 0.021}$ | $\mathbf{0.435}_{\pm 0.013}$ |
| lumo | $0.601_{\pm 0.050}$ | $0.646_{\pm 0.013}$ | $\underline{0.540}_{\pm 0.005}$ | $\mathbf{0.515}_{\pm 0.007}$ |
| gap | $0.630_{\pm 0.044}$ | $0.698_{\pm 0.004}$ | $\underline{0.584}_{\pm 0.007}$ | $\mathbf{0.552}_{\pm 0.008}$ |
| r2 | $0.452_{\pm 0.087}$ | $0.440_{\pm 0.024}$ | $\underline{0.261}_{\pm 0.006}$ | $\mathbf{0.204}_{\pm 0.006}$ |
| zpve | $0.402_{\pm 0.032}$ | $0.410_{\pm 0.004}$ | $\underline{0.328}_{\pm 0.005}$ | $\mathbf{0.284}_{\pm 0.005}$ |
| u0_atom | $0.308_{\pm 0.055}$ | $0.245_{\pm 0.006}$ | $\underline{0.193}_{\pm 0.002}$ | $\mathbf{0.163}_{\pm 0.001}$ |
| cv | $0.291_{\pm 0.118}$ | $0.337_{\pm 0.018}$ | $\underline{0.148}_{\pm 0.004}$ | $\mathbf{0.127}_{\pm 0.002}$ |

Table 3: MSE on the graph regression task. The best results with a statistical significance of $p < 0.05$ are highlighted: the best overall are in bold, the best among pooling methods are underlined.

## B EXPERIMENTAL DETAILS

For the WL kernel, we used the implementation provided in the GraKeL library[2]. The pooling strategy based on Graclus, is taken from the ChebyNets repository[3].

---

[1] http://quantum-machine.org/datasets/
[2] https://ysig.github.io/GraKeL/dev/
[3] https://github.com/mdeff/cnn_graph

## B.1 CLUSTERING ON CITATION NETWORKS

Diffpool and minCUTpool are configured with 16 hidden neurons with linear activations in the MLP and MP layer, respectively used to compute the cluster assignment matrix $\mathbf{S}$. The MP layer used to compute the propagated node features $\mathbf{X}^{(1)}$ uses an ELU activation in both architectures. The learning rate for Adam is 5e-4, and the models are trained for 10000 iterations. The details of the citation networks dataset are reported in Tab. 4.

Table 4: Details of the citation networks datasets

| Dataset | Nodes | Edges | Node features | Node classes |
|---------|-------|-------|---------------|--------------|
| Cora | 2708 | 5429 | 1433 | 7 |
| Citeseer | 3327 | 9228 | 3703 | 6 |
| Pubmed | 19717 | 88651 | 500 | 3 |

## B.2 GRAPH CLASSIFICATION

We train the GNN architectures with Adam, an $L_2$ penalty loss with weight 1e-4, and 16 hidden units ($H$) both in the MLP of minCUTpool and in the internal MP of Diffpool. *Mutagenicity*, *Proteins*, *DD*, *COLLAB*, and *Reddit-2k* are datasets representing real-world graphs and are taken from the repository of benchmark datasets for graph kernels[4]. *Bench-easy* and *Bench-hard*[5] are datasets where the node features $\mathbf{X}$ and the adjacency matrix $\mathbf{A}$ are completely uninformative if considered alone. Hence, algorithms that account only for the node features or the graph structure will fail to classify the graphs. Since *Bench-easy* and *Bench-hard* come with a train/validation/test split, the 10-fold split is not necessary to evaluate the performance. The statistics of all the datasets are reported in Tab. 5.

Table 5: Summary of statistics of the graph classification datasets

| Dataset | samples | classes | avg. nodes | avg. edges | node attr. | node labels |
|---------|---------|---------|------------|------------|------------|-------------|
| Bench-easy | 1800 | 3 | 147.82 | 922.66 | – | yes |
| Bench-hard | 1800 | 3 | 148.32 | 572.32 | – | yes |
| Mutagenicity | 4337 | 2 | 30.32 | 30.77 | – | yes |
| Proteins | 1113 | 2 | 39.06 | 72.82 | 1 | no |
| DD | 1178 | 2 | 284.32 | 715.66 | – | yes |
| COLLAB | 5000 | 3 | 74.49 | 2457.78 | – | no |
| Reddit-2K | 2000 | 2 | 429.63 | 497.75 | – | no |

## C ARCHITECTURES SCHEMATA

Fig. 8 reports the schematic representation of the minCUTpool layer; Fig. 9 the GNN architecture used in the clustering and segmentation tasks; Fig. 10 the GNN architecture used in the graph classification task; Fig. 12 the GNN architecture used in the graph regression task; Fig. 11 the graph autoencoder used in the graph signal reconstruction task.

---

[4]https://ls11-www.cs.tu-dortmund.de/staff/morris/graphkerneldatasets
[5]https://github.com/FilippoMB/Benchmark_dataset_for_graph_classification

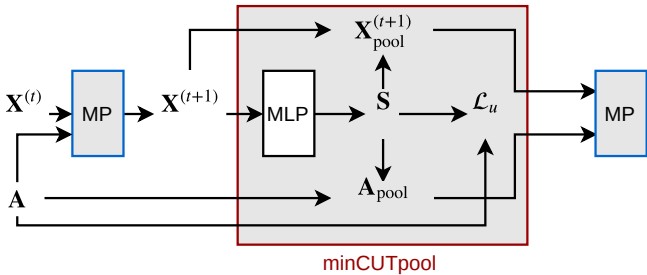

Figure 8: Schema of the minCUTpool layer.

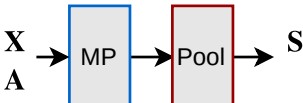

Figure 9: Architecture for clustering/segmentation.

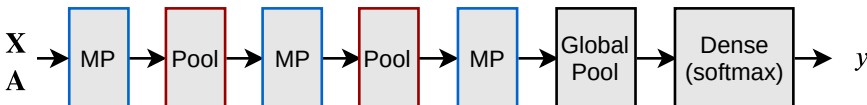

Figure 10: Architecture for graph classification.

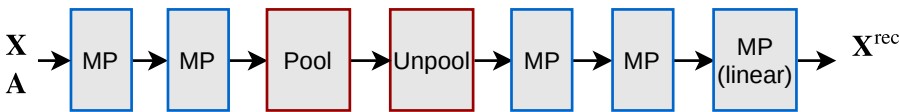

Figure 11: Architecture for the autoencoder.

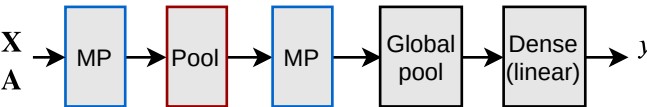

Figure 12: Architecture for graph regression.

