# OpenReview forum: "Mincut Pooling in Graph Neural Networks"
_ICLR.cc/2020/Conference — Reject_

### Official Review · AnonReviewer2 · 2019-10-24
**Official Blind Review #2**

**Rating:** 8

**Review:**

This paper proposes a solution to the important problem of pooling in graph neural networks. The method relies on minimizing a surrogate function inside the standard SGD loop and in conjunction with the optimization of the model parameters - such loss function aiming at optimizing the minCut on the graph. By that it aims to effective achieve a soft clustering of nodes that are both well connected and that have similar embeddings. This in an elegant choice, somewhat resembling the DiffPool method since it's also end-to-end trainable. However it adds the local graph connectivity information due to the minCut loss (and related orthogonality penalty to achieve non trivial solutions on the relaxed minCut continuous problem). Such local graph connectivity is indeed important information to consider when carrying out pooling.
Results show good performance improvement on different tasks of graph clustering, node and whole graph classification. The paper is well written and clear to read. The math is solid and the concept is well substantiated by results.
I found no mention about code release and I would solicit the authors to release the code to reproduce the experiments.

**Experience Assessment:**

I have read many papers in this area.

**Review Assessment: Checking Correctness Of Derivations And Theory:**

I carefully checked the derivations and theory.

**Review Assessment: Checking Correctness Of Experiments:**

I carefully checked the experiments.

**Review Assessment: Thoroughness In Paper Reading:**

I read the paper thoroughly.

---

> ### Author Response · Authors · 2019-11-06
> **Answer to reviewer #2**
>
> We thank the reviewer for their positive assessment of our work and for having carefully read our paper.
>
> We have ready the code of the minCUT pooling layer (both in Pytorch and TF/Keras) to be released on the principal libraries for Graph Neural Networks after the review period is over.
>
> We will also make a Github repo with all the scripts to reproduce the experiments in the paper.
> A preliminary version of the repo can be downloaded from the link on the top of this page.

---

### Official Review · AnonReviewer1 · 2019-10-25
**Official Blind Review #1**

**Rating:** 3

**Review:**

The authors propose a differentiable pooling method for graph data, known as minCUTpool. It learns a clustering assignment matrix using MLPs and then add regularization terms to encourage the clustering results close to the minCUT. The experimental results show that the regularization terms can help improve the performance.

Cons:
1. The novelty is limited. Compared with existing work DiffPool, the proposed method is improving the Diffpool by adding two regularization terms. In addition, the main regularization $L_c$ is already proposed in previous studies.
2. The motivation is not clear. Why should we apply minCut for graph pooling? Intuitively, how is the minCUT related to graph representation learning? The minCut can identify dense graph components but why these dense components should be different clusters in graph pooling? In addition, the author claim “cluster together nodes which have similar features”. How could minCut terms lead to such conclusion?
3.  Important baselines are missing, such as Sortpool (Zhang et al, An end-to-end deep learning architecture for graph classification, AAAI 2018), Self-attention pool (Lee et al, Self-Attention Graph Pooling, ICML 2019).
4. The graph classification results are not convincing enough. In the original Top-K paper (Gao et al , Graph U-Net, ICML2019), the reported results for Proteins and DD datasets are 77.68%, 82.43%, which are significantly better than the results reported in this paper.

**Experience Assessment:**

I have published one or two papers in this area.

**Review Assessment: Checking Correctness Of Derivations And Theory:**

N/A

**Review Assessment: Checking Correctness Of Experiments:**

I assessed the sensibility of the experiments.

**Review Assessment: Thoroughness In Paper Reading:**

I read the paper at least twice and used my best judgement in assessing the paper.

---

> ### Author Response · Authors · 2019-11-06
> **Answer to reviewer #1**
>
> Thanks for your comments. We address them in the following.
>
> 1. Our method and DiffPool perform clustering on the graph vertices. Therefore, they share similarities in expressing the clustering operation through matrix multiplications. However, the outcome is completely different since the two approaches are different in all the constituent parts:
> - computation of the soft cluster assignments;
> - computation of the pooled adjacency matrix is different;
> - unsupervised loss, which is the most crucial component and was carefully derived from theoretical principles.
> We spent a huge amount of space in the paper to explain those differences and to show that our method yields results that are quantitatively better (time/accuracy) and qualitatively different.
>
> Naturally, $L_c$ is the term that quantifies the minimum cut and has been an important research subject in the past decades, but it has never been used to design GNN pooling operators. As discussed in sec 2, $L_c$ alone cannot be minimized directly, since is non-convex and reaching the minimum is not guaranteed without additional constraints. In fact, most of the research efforts in the spectral clustering field focused on formulating such constraints.
>
> In this paper, we leverage the mechanisms of the GNN to obtain a good initial estimate of the cluster assignments from the node features. Indeed, the graph convolutions make the features of strongly connected vertices on the graph similar and, since the MLP is a smooth function, it generates similar cluster assignments for such vertices.
> Then, we formulate a soft constraint, $L_o$, which is cheap to compute compared to other orthogonality constraints proposed in the literature.
>
>
> 2. Let us use an analogy with CNNs used in computer vision, which is easier to understand also for readers not familiar with GNN pooling.
> CNNs exploit the assumption that neighbouring pixels are strongly related and they can be pooled together by extracting local summaries. Similarly, the strongly connected components on a graph are highly related and their features become more and more similar after MP. Therefore, it is reasonable to design a pooling operation that extracts local summaries by clustering together such components and reduce the graph accordingly. In this way, the next MP will exchange information between parts of the graph, which were originally weakly connected.
> Thus, pooling helps to gradually distil global information from the graph, such as the class label, by generating a hierarchy of coarsened representations of the graph.
>
> The MLP assigns nodes with similar features to similar clusters because the MLP is a smooth function. The smoothness property guarantees that the function, when fed with similar inputs, yield similar outputs.
>
>
> 3. We are well aware of both works. In Sortpool there is not an unsupervised loss and, therefore, it is not possible to use it in clustering and segmentation tasks in 5.1. About the graph classification, our study focuses on developing a deep GNN architecture, where MP layers are alternated with pooling layers. In the Sortpool architecture pooling is performed just one time and, afterwards, it is not possible to perform MP anymore since Sortpool does not generate a coarsened graph.
>
> SAG pool only differs from Top-K in the way it computes the scores y. While in Top-K $y = Xp/|p|$, in SAG $y = \sigma(D^{-1/2} AD^{-1/2} Xp)$. In other words, SAG just applies an additional MP to the node features X. Since we already apply a MP before pooling, the node features used to compute y in Top -K are already propagated.
>
>
> 4. The results obtained by Top-K are reproducible with public implementations in GNNs libraries and with the provided code.
>
> The Graph U-net architecture is significantly different from the GNN used in our experiments:
> - their model has higher capacity: 8 trainable MP layers, compared to 3 in our setting;
> - It uses skip connections at each block.
>
> We believe that the skip connections conceal the true effect of Top-K pooling since they allow to gather all the information just from the first layer. Interestingly, also other works that adopt Top-K (e.g., Cangea at al. “Towards sparse hierarchical graph classifiers”) use skip connections. However, these works do not compare with a flat baseline making impossible to evaluate the improvements of pooling.
>
> One of our findings is that Top-K is, in fact, detrimental since the results of the flat baseline deteriorate when Top-K is inserted. The reason is that Top-K drops entire parts of the graph: in the analogy with CNNs, it would be like if pooling drops either the left or the right part of an image, making the classification impossible if both parts are necessary for classification. This conclusion is also supported by other recent works (Knyazev et al., "Understanding Attention and Generalization in Graph Neural Networks", 2019) and by the qualitative analysis performed with an AutoEncoder in section A.1 of our supplementary material.

---

> > ### Author Response · Authors · 2019-11-14
> > **SAGPool results**
> >
> > Dear reviewer,
> > we have included the results from SAGPool in the graph classification experiments. The results are reported in Tab. 2 in the revised version of the paper and we have also included the SAGPool layer in the code provided with our submission.
> >
> > However, we would like to comment that the differences with Top-K are not statistically significant. We report in the bottom the accuracy and standard deviations on 10 runs obtained by the two methods. A significant difference is for p-values < 0.05 or 0.01, but in this case the p-values are much higher than that.
> >
> > +-------------------+-------------------- -------+---------------------------+------------+
> > |                         | TopK                          | SAG-pool                  |                |
> > +-------------------+----------------------------+---------------------------+------------+
> > |                         | Acc mean | Acc std | Acc mean | Acc std | p-value |
> > +-------------------+---------------+------------+--------------+-----------+------------+
> > | Bench-easy   | 82.4           | 8.9          | 84.2          | 2.3        | 0.5435   |
> > +-------------------+---------------+------------+--------------+-----------+------------+
> > | Bench-hard   | 42.7           | 15.2       | 37.7          | 14.5       | 0.4614  |
> > +-------------------+---------------+------------+--------------+-----------+------------+
> > | Mutagenicity | 71.9           | 3.7         | 72.4          | 2.4         | 0.7241  |
> > +--------------------+--------------+------------+--------------+-----------+------------+
> > | Proteins          | 69.6          | 3.5          | 70.5          | 2.6        | 0.5222   |
> > +--------------------+--------------+------------+--------------+-----------+------------+
> > | DD                    | 69.4          | 7.8         | 71.5          | 4.5         | 0.4704  |
> > +--------------------+--------------+------------+--------------+-----------+------------+
> > | COLLAB           | 79.3          | 1.8         | 79.2          | 2            | 0.9077  |
> > +--------------------+--------------+------------+--------------+-----------+------------+
> > | Reddit-Binary | 74.7          | 4.5         | 73.9          | 5.1         | 0.7143  |
> > +--------------------+--------------+------------+--------------+-----------+------------+

---

### Official Review · AnonReviewer4 · 2019-11-01
**Official Blind Review #4**

**Rating:** 3

**Review:**

This paper proposes a graph pooling method by utilizing the Mincut regularization loss. It is an interesting idea and performs well in a number of tasks. However, due to the limitation of novelty and poor organizations, this paper cannot meet the standard of ICLR. The detailed reasons why I give a weak reject are listed as follows:

1. Even though the proposed minCUT pool is interesting, the contribution is not enough to get published in the ICLR. If I understand correctly, the only difference is the unsupervised loss, compared with the previous work, Diffpool [1].

2. The paper needs to be reorganized to demonstrate its contribution. The proposed method section only has around 1.5 pages, making it difficult to understand the proposed method clearly. Therefore, more details and analyses about the proposed method should be included to support and clarify the idea.

3. The paper needs to be improved for its theoretical derivations and proof. For example, it is not clear why Equation (6) is correct, which is the main contribution of this paper. The authors provide intuitive thoughts but there are not theoretical derivations and proof. The term $L_c$ comes from Equation (2) but why is it correct to only compute the trace?

4. Some experiments cannot support the claim very well.  For example, the graph clustering experiments are not convincing. The goal of graph pooling is to learn high-level graph embeddings but not perform graph clustering. It is not proper to evaluate the graph pooling method using graph clustering tasks. Or, the author should clarify the motivation to do this experiment. If the model is trained for graph classification or node classification, then why should the node clusters lead to high NMI or CS scores?

[1]. Ying et al., Hierarchical Graph Representation Learning with Differentiable Pooling, NIPS 2018


==========Update===========

I have read authors response and other reviews. While the authors address some of my concerns, I still believe the contribution/novelty is limited. I am sticking to my score.

**Experience Assessment:**

I have published one or two papers in this area.

**Review Assessment: Checking Correctness Of Derivations And Theory:**

N/A

**Review Assessment: Checking Correctness Of Experiments:**

I assessed the sensibility of the experiments.

**Review Assessment: Thoroughness In Paper Reading:**

I read the paper at least twice and used my best judgement in assessing the paper.

---

> ### Author Response · Authors · 2019-11-06
> **Answer to reviewer #4**
>
> Thanks for your comments. We reply to each point in the following.
>
> 1. Both our method and Diffpool cluster the vertices of an annotated graph, but their outcomes are completely different since the two approaches are different in all the constituent parts:
> - computation of the soft cluster assignments
> - computation of the pooled adjacency matrix
> - clustering objective (unsupervised loss)
> We spent a huge amount of space in the paper to explain those differences and to show that our method yields results that are quantitatively better (in time and accuracy) and qualitatively different.
> We explain the results by revealing serious flaws in existing differentiable pooling methods (Diffpool cannot even partition a regular grid properly), since they are based on heuristics rather than on theoretically grounded principles, like our method.
>
>
> 2. We understand that our paper assumes the readers have some familiarity with pooling in GNNs and related works. However, we would like to kindly ask the reviewer how they think the paper should be reorganized.
>
> At the moment, in Section 1 we introduce the problem of pooling in GNNs, anticipating the problems and limitations of existing approaches. Section 2 provides all the theoretical backgrounds and the math necessary to understand the new methodology, which is presented in Section 3.
> Then, in Section 4 we review in detail the principal methods in the GNN pooling literature, underlining their drawbacks, which are addressed by our approach.
> In Sec. 5 we perform a detailed analysis of the proposed method, to support our claims, clarify its mechanisms, and demonstrate how and why it outperforms competing methods in several ways.
> Finally, the supplementary material provides additional experiments and analyses, which can help the interested reader to acquire additional insights on the behaviour of the different pooling methods.
>
>
> 3.  Eq. 6 is correct because the two quantities $\sum_k C^T_k A C_k$ and $tr(C^TAC)$ are identical according to basic properties of the trace operator in linear algebra. The latter is only more convenient to implement in those software libraries supporting vectorization (PyTorch, TensorFlow, etc..).
>
> We believe that all the elements in the methodology section are either supported by the theory and the references provided in Section 2 or are carefully discussed and explained in Section 3, and illustrated through experiments and visualization in Section 5.
>
> Therefore, we would like to kindly ask the reviewer: which is the part in the proposed methodology that lacks a theoretical derivation and would require a proof?
>
>
> 4. Neural networks used in computer vision exploit the assumption that neighbouring pixels are strongly related and they can be pooled together by extracting local summaries. Similarly, the strongly connected components on a graph are highly related and their features become more and more similar after performing message-passing. Therefore, it is reasonable to design a pooling operation that extracts local summaries by clustering together such components and reduces the graph accordingly, even if we agree that clustering is not the PRIMARY purpose of graph pooling.
>
> We believe that it is proper to evaluate on clustering tasks those pooling methods that are based on clustering (SC, DiffPool, and minCUT).
> Therefore, the experiments in Section 5.1 show that the proposed pooling method naturally provides the GNN with a coarsened representation of the graph that is quantifiably good. Compared to DiffPool on the same tasks, we show that DiffPool is actually introducing noise in the learning process because it leads to poor pooling results when used as a stand-alone component.
> We use NMI and CS to evaluate the pooling performance of different methods on this task because they are metrics that are often considered to assess the quality, in terms of purity, of a clustering.
>
> As explained at the beginning of 5.1, the clustering/segmentation experiments are performed in a completely unsupervised setting, i.e., in the absence of a supervised loss, such as cross-entropy used in graph classification. This means that the network used in sec 5.1 is NOT trained for graph or node classification, and there are no labels involved in the training.
>
> We clarified better the purpose of these experiments in Sec 5.1.

---

### Author Response · Authors · 2019-11-10
**Revised version of the paper uploaded**

Dear reviewers,

we have uploaded a revised version of the manuscript (and of the code), which addresses the requests and the comments of the reviewers.
Together with our answers, we hope we have clarified all the doubts and concerns of the reviewers.

---

### Decision · Program_Chairs · 2019-12-19

**Decision:**

Reject

**Comment:**

Two reviewers are negative on this paper while the other reviewer is positive. Overall, the paper does not make the bar of ICLR. A reject is recommended.